# Continuous Locomotion Mode and Task Identification for an Assistive Exoskeleton Based on Neuromuscular–Mechanical Fusion

**DOI:** 10.3390/bioengineering11020150

**Published:** 2024-02-02

**Authors:** Yao Liu, Chunjie Chen, Zhuo Wang, Yongtang Tian, Sheng Wang, Yang Xiao, Fangliang Yang, Xinyu Wu

**Affiliations:** 1Shenzhen Institute of Advanced Technology, Chinese Academy of Sciences, Shenzhen 518055, China; ll.liu1@siat.ac.cn (Y.L.); zhuo.wang@siat.ac.cn (Z.W.); yt.tian@siat.ac.cn (Y.T.); xiaoyang@siat.ac.cn (Y.X.); fl.yang@siat.ac.cn (F.Y.); xy.wu@siat.ac.cn (X.W.); 2Guangdong Provincial Key Lab of Robotics and Intelligent System, Shenzhen Institute of Advanced Technology, Chinese Academy of Sciences, Shenzhen 518055, China; s.wang3@siat.ac.cn; 3Guangdong-Hong Kong-Macao Joint Laboratory of Human-Machine Intelligence-Synergy Systems, Shenzhen Institute of Advanced Technology, Chinese Academy of Sciences, Shenzhen 518055, China

**Keywords:** assistive exoskeleton, neuromuscular–mechanical, locomotion modes and tasks, human–machine collaboration

## Abstract

Human walking parameters exhibit significant variability depending on the terrain, speed, and load. Assistive exoskeletons currently focus on the recognition of locomotion terrain, ignoring the identification of locomotion tasks, which are also essential for control strategies. The aim of this study was to develop an interface for locomotion mode and task identification based on a neuromuscular–mechanical fusion algorithm. The modes of level and incline and tasks of speed and load were explored, and seven able-bodied participants were recruited. A continuous stream of assistive decisions supporting timely exoskeleton control was achieved according to the classification of locomotion. We investigated the optimal algorithm, feature set, window increment, window length, and robustness for precise identification and synchronization between exoskeleton assistive force and human limb movements (human–machine collaboration). The best recognition results were obtained when using a support vector machine, a root mean square/waveform length/acceleration feature set, a window length of 170, and a window increment of 20. The average identification accuracy reached 98.7% ± 1.3%. These results suggest that the surface electromyography–acceleration can be effectively used for locomotion mode and task identification. This study contributes to the development of locomotion mode and task recognition as well as exoskeleton control for seamless transitions.

## 1. Introduction

Wearable robotic devices—both exosuits and exoskeletons—show potential benefits in assisting humans in walking, reducing metabolism, and enhancing endurance. The design concept of wearable robotic devices is to provide appropriate joint torques and timing based on biomechanical gait analysis [1,2]. Both passive and active devices can contribute to a reduction in energy consumption.

Along with the advantages of exoskeletons comes the weight of systems and constraints on the human body, which results in uncomfortable sensations. Therefore, researchers are interested in exploring additional methods to significantly reduce energy expenditure to compensate for negative influences. In addition to structural improvement, control strategies have been a hot topic of research, essentially focusing on two areas: (1) personalized optimization using human-in-the-loop optimization algorithms [3,4,5] and (2) muscle dynamics [6], which are employed to adjust control parameters. Since the dynamics of human joints vary with the mode of locomotion, e.g., walking on level ground, a slope, or stair ascent/descent [7,8,9], accurate locomotion mode identification and real-time control strategies pose challenges with respect to the use of wearable devices.

Multiple sensors are applied to locomotion mode recognition. Sensors currently used for exoskeleton pattern detection systems can be categorized into three major groups: mechanical, biosignal, and vision sensors.

Many different strategies have been employed using mechanical sensors, such as the use of load cells [10] and inertial measurement units (IMUs) [11,12]. These methods predict locomotion modes by threshold or machine learning according to a human state, such as limb motions or heel contact forces [13]. Mechanical sensors are characterized by convenient, low cost, and relatively good recognition performance. However, mechanical sensors have a step delay, leading to uncomfortable sensations and increased risks of falls. Surface electromyography (sEMG) represents muscle activation and is widely applied in wearable devices for intention recognition [14]. The features of sEMG in the time or frequency domain are often extracted and input to several pretrained classifiers for mode identification. Compared to the responses of mechanical sensors to movements, sEMG signals precede limb motion by 20–150 ms [7,15], which can compensate for the delay of the system. sEMG is also associated with uncertainties, such as those relating to the electrode position, crosstalk, and noise, so measurements must be taken from multiple muscles to ensure a reasonable recognition rate. At present, the combination of mechanical and biosignal sensors is the most widely used method to improve the accuracy of recognition. Studies have shown that the fusion of mechanical and biosignal sensors can significantly improve the classification of gait modes [7,16].

Recently, the use of vision sensors has been proposed for the identification of terrain. Depth images are employed to detect user movement intent or to recognize real-life environmental context through edge detection or 3D point cloud classification [17,18]. The authors of a previous study achieved reliable confidence estimation, reporting a decision result 0.6 to 4 s ahead of the motion intention [19]. Vision sensors are generally large, and this method requires a great deal of computational power. In addition, visual sensors can only be used to recognize terrain (such as level ground, slopes, and stairs) and cannot consider the impact of tasks (such as speed and load) on assistive strategies.

Many experimental studies have pointed out that both kinematics and kinetics change considerably as speed and load change [9,20,21]. Therefore, it is critical to adopt different control strategies for corresponding tasks on specific terrain types. Spatiotemporal gait parameters during walking have been extensively studied in both healthy and pathological populations [22]. Walking speed is one of the most important parameters characterizing people’s daily mobility [23] and has been recognized as a proxy measure of ambulation quality. Consequently, speed recognition is a crucial parameter with respect to control strategies for assistive exoskeletons. Gait speed estimation is widely used in clinical applications to analyze the characteristics of walking-related diseases. Another central application area for speed estimation is in sports, where devices like smart watches are employed to predict the speeds for professional athletes or runners. The use of global navigation satellite systems is a highly accurate method for speed measurement [24]. These systems need to be used outside and away from tall buildings. Furthermore, communicating with satellites consumes a considerable amount of power, limiting operating time. Due to the low power consumption and portable characteristics of electrical sensors, inertial sensor-based methods have been developed for speed estimation [22,25]. Algorithms using inertial sensors to estimate walking speed can be grouped into two categories: black-box models based on machine learning [26] and human gait models [23].

Similar to walking speed, it has been demonstrated that carrying loads significantly affect human gait biomechanics [27,28]. Much of the current load carriage research has concentrated on the fundamental tasks of military personnel. Previous research has shown that carrying loads can impact the spatiotemporal parameters of human gait and influence the efficiency and safety of movement, increasing the risk of musculoskeletal injury [29,30]. Although the physiology and biomechanics of load-bearing carriage properties have been characterized [31], few teams have studied the identification of carriage weights for assistive exoskeletons. Weight can be measured initially, but such measurements are impossible during a mission, such as a march. Therefore, it is necessary to recognize load carriage in real time and provide an appropriate control strategy.

In order to assist human walking more efficiently and naturally, the exoskeleton must consider the modes and tasks of gait in the control strategies. An intuitive method is to utilize different control strategies for different gait movements. However, human walking parameters vary significantly based on terrains, speeds, and loads. Up to this point, the primary focus of most exoskeletons has been on terrain recognition [32]. This is mainly because the open-source dataset of human lower limb biomechanics emphasizes gait analysis [8,9] rather than the control strategies of exoskeletons. To our best knowledge, only one study has addressed the effect of walking tasks [6], but it only focused on speeds.

Accordingly, the objective of this study was to achieve the continuous recognition of locomotion modes (level/incline) and tasks (speed/load) through the fusion of neuromuscular–mechanical data for assistive exoskeletons. We formulated two hypotheses: (1) that combining accelerations (ACCs) and sEMG signals would prove effective for recognizing both mode and task, and (2) that incorporating a greater number of features would lead to higher estimation accuracy. To achieve this goal, we gathered a comprehensive dataset of human gait data from injury-free adult participants across various locomotion modes and tasks. This dataset served as the foundation for investigating the optimal algorithm, feature combinations, window length, and window increment. Furthermore, we conducted an assessment of the algorithm’s robustness, considering practical factors.

## 2. Materials and Methods

### 2.1. Participants and Protocol

Seven healthy volunteers (age: 24 ± 2, weight: 67 ± 6 kg, height: 175 ± 6 cm) without any physical disease were recruited for this experiment. All participants were healthy and injury free to both lower and upper limbs in the last six months.

Before the start of experiments, all participants were asked to sign an informed consent form. The experimental procedures, specifications, and risks were comprehensively explained to each participant. The experiments were conducted under the protocol of the ethics committee of Shenzhen Institute of Advanced Technology, Chinese Academy of Sciences (SIAT-IRB-221215-H0632). Anthropometric and demographic information of the participants is shown in Table 1.

This study was conducted in low-power medical VLSI and Body Sensor Networks Laboratory Key Lab for Biomedical Informatics and Health Engineering, Chinese Academy Sciences. Accelerations in three axes, sEMGs signals from eight muscles, and ground reaction forces were recorded synchronously in real time. The acceleration and sEMG signals were captured using a 16-channel wireless EMG system sampled at 2000 Hz (Delsys Trigno^®^ 1.2, Natick, MA, USA). Ground reaction forces were recorded using an instrumental treadmill sampled at 1000 Hz (AMTI, MA, USA). Eight muscles of the right leg were selected: *gluteus maximus* (GM), *rectus femoris* (RF), *vastus lateralis* (VL), *vastus medialis* (VM), *biceps femoris* (BF), *tibialis anterior* (TA), *gastrocnemius medialis* (GAS), and *Soleus* (SOL). The selection of muscles referred to results of Ding [33].

Six types of patterns were investigated: level ground walking at different speeds (0.75 m/s, 1.25 m/s, 2 m/s, and 2.75 m/s), 15% incline walking at 1.25 m/s, and 10 kg load walking at 1.25 m/s. An AMTI treadmill was used to achieve changes in speed and incline, and the load vest was used to achieve load changes. First, an experienced rehabilitation therapist performed the measurement and sensor-affixing work. Participants were asked to wear swimming trunks and sneakers. A tape measure was utilized to measure the leg length (from the anterior superior iliac spine to the medial malleolus) and body height. The weight was measured by a mass scale. The areas where electrodes were attached were wiped with alcohol-soaked cotton swabs, and any long body hair was trimmed. The EMG electrodes were pasted by double-side adhesive and secured by self-adhesive banding, as shown in Figure 1. The sEMG acquisition followed the SENIAM guidelines [34]. Then, the adjustment process was conducted. Patients adapted to different speeds and weight loads. When the patient felt ready for the experiment, the above-mentioned patterns were collected sequentially. The locomotion modes and tasks listed above were collected in sequence. Each pattern recorded at least 5 trials and each trial included at least 7 gait cycles. A gait cycle was defined from one heel strike of a foot to the next heel strike of the same foot. The first point through which the ground reaction force was greater than 15 N was regarded as the beginning of a gait cycle. There was a 2-min break between each pattern to prevent muscle fatigue.

### 2.2. Data Processing

This study aimed to design a method for continuously estimating locomotion modes and tasks, generating a decision stream for exoskeleton to apply control strategy. The system structure is depicted in a block diagram shown in Figure 1. The neuromuscular–mechanical fusion signals acquired simultaneously were streamed into a preprocessing session to remove the noise. Feature extraction was applied to the preprocessed signals to obtain a multimodal feature matrix. The matrices of sEMG and ACCs were then downsampled to match the length of GRF. Finally, the post-processing stage was conducted to obtain continuous control decisions and combine the classification results with the gait phase detector.

It is inevitable that unwanted noises would be produced during locomotion, which must be removed. Raw sEMG signals were filtered by a sixth-order bandpass Butterworth filter with cutoff frequencies of 15–500 Hz to remove electrical noise and motion artifacts. The mean value of the corresponding axis was subtracted from all ACC signals to reduce the direct current offset. Raw GRF signals were filtered by a fourth-order low-pass Butterworth filter with a cutoff frequency of 12 Hz. The zero-lag algorithm was applied to all filters to avoid phase shifts. All signals were then segmented by sliding analysis windows for continuous classification decision making. The length of the analysis window was initially set to 140 (equivalent to 70 ms) with a window increment of 10 (equivalent to 5 ms).

The preprocessed signals were then subjected to the feature extraction process. The features of sEMG and the mean value of ACCs were computed within an analysis window mentioned above. The preprocessed GRF signals were passed into the gait phase detector to segment gait cycles and define phases. A feature matrix of EMG and ACCs signals was finally formulated. The feature matrix was downsampled by a factor of two to match the length of GRF signals. The downsampled feature matrix was normalized from 0 to 1 to improve the performance of classifiers. Four gait cycles were used for the training set, and the final gait cycle was used for the test set. Support Vector Machine (SVM) [35], K-Nearest Neighbor (KNN) [36], Quotient-Difference Algorithm (QDA) [37], and Decision Tree (DT) [38] were verified successively. Cross-validation was conducted for each classifier ahead of classification.

### 2.3. Parameters on Performance

*Classifier*. The performance of locomotion mode estimation has been reported with different classifiers primarily including SVM, LDA, and ANN [7,14]. This study explored more classifiers for higher performance in locomotion mode and task estimation. We aimed to implemented functionality to reduce the latency of the recognition system. Furthermore, a total of four classifiers were explored, as mentioned earlier. Support Vector Machine (SVM), K-Nearest Neighbor (KNN), Quadratic Discriminant Analysis (QDA), and Decision Tree (DT) are all popular machine learning algorithms used for classification tasks. These algorithms work by analyzing the patterns in the training data and then using that information to make predictions for new, unseen data. In comparison to complex algorithms, these classifiers are known for their fast operation speed and high precision. The classifier of SVM was implemented using libsvm 3.3. The statistics and machine learning toolbox 12.0 [39] was used for the remaining classifiers. Since using more features results in longer running times, only RW, WL, and ACCs were selected for this process. A detailed introduction to the algorithms will not be provided in this paper.

*EMG features*. The selected features are closely related to the accuracy of estimation. Furthermore, the success of the recognition system depends on whether the selected features effectively reflect the problem under study. Many researchers have successfully achieved terrain estimation using root mean square (RMS), mean absolute value (MAV), number of zero crossings (ZC), waveform length (WL), and number of slope sign changes (SSC). Few researchers explored the individual contribution of each feature and the optimal combinations. In this study, we investigated a total of 15 individual features or combinations thereof. The confusion matrices of a signal feature and combinations were calculated. Table 2 displays the feature sets used in the estimation and abbreviations for feature combinations. The formulas for feature extraction are provided in Appendix A. Additionally, the Kruskal–Wallis one-way analysis of variance by ranks was employed to assess the improvement of ACCs. The reason for choosing Kruskal–Wallis was that we found that RMS, MAV, WL, and other features did not follow a normal distribution.

*Length of the analysis window and increment size*. The values of sEMG features would vary with the length of the analysis window. Furthermore, since sEMG is characterized as a broad–sense Gaussian random process [40], the window length should not exceed 200 ms to minimize variation [7]. To ensure synchronization between exoskeleton assistive force and human limb movement, we investigated window lengths ranging from 30 (15 ms) to 200 (100 ms) data points. The selected maximum window length was due to the fact that one gait cycle takes less than 160 ms when running at 2.75 m/s, and there were inevitable system delays. Additionally, sEMG signals typically precede limb movements by 30–150 ms; therefore, the window length can impact human–machine collaboration and should be adjusted considering system delay. We employed an overlapping windowing scheme to facilitate a quick decision making, improve signal utilization, and reduce abrupt signal jumps. Smaller window increments result in more decision points within a gait cycle, aiming to achieve smoother control. We compared classification errors and decision points across window increments ranging from 10 to 60 data points. The classification errors were determined by comparing the predicted class label of the final gait to its actual class label. In addition, the Kruskal–Wallis one-way analysis of variance by ranks was used to test window length and increment, respectively.

*The robustness*. The number of muscles greatly affects the practicability of the method used in real life. To better utilize exoskeleton scenes, the robustness of the SVM model was investigated. Muscles were separated into six groups according to function: gluteus, thigh, calf, anterior thigh (AT), posterior calf (PC), and all muscles. The muscles contained in each group are shown in Table 3.

## 3. Results

Both the classifier and the design parameters had a substantial impact on recognition accuracy. Figure 2 displays the classification performance of classifiers using an RMS, WL, and ACCs feature set with an analysis window of 140 and an increment of 10. SVM exhibited the lowest classification error, followed by KNN, QDA, and DT, with average error rates of 1.68%, 2.19%, 3.09%, and 6.96%, respectively. However, classifier performance was inconsistent and varied among participants. For instance, in the case of P2, P4, P6, and P7, KNN demonstrated better estimation accuracy than SVM. Among the results for P5, SVM was the sole classifier to achieve a superior result of 96.94%. Given that SVM maintained consistent accuracy across all participants’ tests, ranging form 95.46% to 99.75%, the SVM classifier was selected for the remainder of this study.

Figure 3 shows the influence of feature and feature combinations on classification accuracy when the SVM algorithm was employed to classify eight channels of data with 140 length of analysis window and 10 window increments. The definitions of each of these features are shown in Table 2. Regarding single signal features, WL achieved the highest accuracy in mode and task recognition (92.51%), while SSC had the lowest accuracy (67.63%). It is worth noting that RMS, MAV, and WL all had outliers at 25.73%, 25.90%, and 80.13%, respectively, all from P7. The ACCs feature also yielded relatively good results with an accuracy of 93.09%. The combination of features improved the classification accuracy. Interestingly, the accuracy of all feature combinations (five features of EMG and ACCs) was not the best (98.57%). The best results were obtained with the feature set of RMS, WL, and ACCs (98.32%). In other words, adding features may potentially decrease estimation accuracy. The accuracies of R/W, R/M/W, R/M/W/S feature set were 93.20%, 92.81%, and 91.63%, respectively. The impact of ACCs on the classification accuracy is detailed in Table 4. No significant differences were found between non-ACCs and ACCs feature sets. Overall, ACCs features improved accuracy except for R/W/A of P3, R/M/W/A of P1, and P3. Considering that the feature matrix of E/A had 64 dimensional while R/W/A had 40, and the difference in accuracy between the two sets was small (0.25%), R/W/A was selected for the algorithm, balancing response time and accuracy.

The effects of window increment are displayed in Figure 4 when the SVM algorithm was utilized to classify eight channels of data using a 200 ms window length and R/W/A feature set. Window increments had a minimal effect on the classification accuracy. The highest accuracy achieved was 99.25% when the increment was 60, while the lowest was 98.03% when the increment was 40. The most significant impact of window increments was on the number of decisions made, which directly affected the system’s decision delay. Decision points were counted at the highest speed for each participant within a gait cycle. Furthermore, the system selected the minimum decision count among all participants, which corresponded to the maximum decision delay, as displayed in Table 5. This process ensured that the system worked for everyone. Since there were also hardware delays, such as data transfer delays, the system needed to ensure that the decision delay did not exceed 15 ms. Considering the system delay, a window increment with higher accuracy (increment = 20) was chosen for the algorithm.

Figure 4 shows the effects of window length on classification accuracy. As the length decreased, the estimation accuracy generally showed a downward trend. The classifier produced favorable results within the window length range of 90 to 200 with the highest performance observed at a length of 170, reaching 98.79%. A linear correlation between accuracy and window length was observed in the range of 30 to 90. Table 6 contains the classification accuracy for each participant. The reduction in features and the enhancement of computer computing power minimized the impact of changes in window length on decision delay. Furthermore, there were no significant differences observed between window lengths.

The exploration of SVM algorithm robustness was performed as a separate step with a window length of 140, an increment of 10, and the R/W/A feature set. For this process, participants P1, P2, P4, and P5 were selected, and the statistical results are depicted in Figure 5. The anterior thigh muscles, namely RF, VL, and VM, achieved the highest accuracy of 94.95%, while Glu obtained the lowest accuracy at 70.53%. When considering only the posterior calf muscles, GAS and SOL, the accuracy was 87.08%. Upon adding TA, the accuracy improved to 91.61%. However, this improvement was not consistent for everyone, as P2’s accuracy increased from 94.21% to 97.71%.

According to the results, the SVM classification algorithm was selected for locomotion mode and speed estimation with the R/W/A feature set, a window length of 170, and an increment of 20. The confusion matrices for R/W/A, ACCs, and R/W are displayed in Figure 6. Analysis of the confusion matrix reveals that acceleration signals were advantageous in recognizing speeds, but they struggled to identify the load task accurately. Recognition accuracy for a 10 kg load and a speed of 1.25 m/s reached 87.4% and 83.4%, respectively. Within the R/W feature set, the lowest recognition accuracy was recorded at 66.0% for a speed of 0.75 m/s. In the R/W/A feature set, all tasks achieved commendable recognition rates ranging from 97.4% to 99.9%.

Figure 7 shows an instance of the identification and decision-making results for P1. The lines represent the desired assistive torque that the ankle exoskeleton should supply, while the dots represent the decision streams made by the system. The assistive torque functions were acquired by fitting 0.2 times the ankle biological torque curves in Gaussian form. The results revealed that the minimum number of decisions reached 32 at a speed of 2.75 m/s. This indicates that the system could provide adequate decisions for continuous control.

## 4. Discussion

The flexibility of exoskeletons in real-life scenarios is limited, which is primarily due to various environmental contexts and the complexity of human locomotion tasks. The most decisive factor in determining the operation mode of gait assistance is generally believed to be the terrain, including level walking, stair climbing, and ramp ascent [41]. However, human walking parameters also exhibit significant variability in different motor tasks, which many studies tend to overlook. In this study, an interface based on sEMG-ACCs signal fusion was developed, which could accurately classify a variety of locomotion modes (level/incline) and tasks (speed/load). Furthermore, the model exhibited high robustness and could still yield favorable results even with only three thigh muscles. Generally, joint sensors and IMUs data are processed by a machine learning or threshold-based method to recognize the situation [42]. This study adopted a fusion method of sEMG and acceleration signals, achieving accurate results in locomotion modes and tasks. By incorporating the biosensor, we eliminated the one-step delay [32]. Compared to the fusion of sEMG and ground reaction forces used by Huang [7], Kyeong [15], and others, the utilization of sEMG and acceleration signals improved the range and accuracy of recognition. The phase-dependent classifiers, as adopted by Kyeong in their study, were not employed in this research for two primary reasons. Firstly, the sEMG signals in a gait cycle exhibited considerable variation, which may lead to feature overlap and, consequently, reduced recognition accuracy. The phase-dependent method was originally designed to address this issue. Secondly, this method reduced multiple decision points and led to decision delays. The SVM classifier produced better classification accuracy, which aligns with the findings of Huang [7] and Kyeong [15]. The reason for this difference may be the varying sensitivity of algorithms to signal quality.

The study has demonstrated that recognition accuracy was not directly proportional to the quantity of features. Regarding signal features, WL achieved the best results, which was possibly because this feature provided information on waveform complexity, aligning with Kyeong’s findings [14]. However, the addition of RMS, MAV, and SCC reduced recognition accuracy. Such examples also existed in R/W/A and E/A, WL and EMG, etc. This may be because the increase in features resulted in overlaps between classes. Therefore, it is necessary to select the combination of features based on the specific problem, and in this study, the RMS, WL, and ACC were extracted. ACCs, as shown in Table 4, have generally improved recognition accuracy, although there was no significant difference. The confusion matrix revealed that ACCs had an advantage in recognizing speed but struggled with load recognition. Figure 6 illustrates that sEMG signals exhibited low recognition accuracy at low speeds, reaching only 66% at 0.75 m/s. This phenomenon arises from the reduced muscle activity at lower speeds, highlighting the significance of incorporating acceleration.

Discrepancies between individual and overall results were present in all investigations. There are two main reasons for this phenomenon: one is individual differences such as walking habits, muscle strength, etc., and the other is electrode position. The first reason is inevitable, so although the overall results improved, there was no significant difference, as indicated in Table 4, Table 5 and Table 6. The window increment mainly impacts the number of decision points (corresponding to the time between two decision points) within a gait cycle, and its determination should be based on specific requirements. A decision delay of 12 ms was chosen due to hardware latency. The window length corresponds to the time difference between the decision number point and muscle activation. Regarding model robustness, the thigh muscles yielded superior results compared to the calf muscles. Task recognition could be accomplished using three muscles on the anterior thigh.

While the method designed based on sEMG–ACCs signal fusion showed significant potential for continuous locomotion mode and task identification, this study also had several limitations. Firstly, this work represents an initial study, which investigated six types of locomotion modes and tasks, including level walking, slope, speed, and load. Secondly, we have only explored recognition and have not yet integrated it with exoskeletons. Thirdly, parameters were obtained based on offline data, and, in particular, the use, window increment, and length need to be re-evaluated for system performance. The aim of this study was to propose a suitable method that can help address task recognition, which is a commonly overlooked aspect in assistive exoskeletons. In future work, it will be essential to quantify varying speeds and loads as participants walk in realistic environments. Additionally, the recognition system would be integrated with the exoskeletons to develop appropriate control strategies. In the next phase of the study, it will be essential to comprehend the assist moment for different types of exoskeletons based on gait phases.

## 5. Conclusions

This study demonstrated an initial attempt to develop and evaluate a neuromuscular–mechanical fusion-based algorithm for identifying locomotion mode and task. Compared to traditional signal fusion, this study introduced acceleration to the locomotion task recognition. The fusion of sEMG and ACCs achieved the identification of speed and load, which are often overlooked in assistive exoskeletons but are very important. The optimal algorithm, feature combination, window length, window increment, and robustness were explored. The recognition system based on the SVM classifier, R/W/A feature set, 170 window length, and 20 increment produced over 98.7% accuracy for recognizing locomotion modes and tasks. These promising results may assist in applying assistive exoskeletons in real-world environments.

## Figures and Tables

**Figure 1 bioengineering-11-00150-f001:**
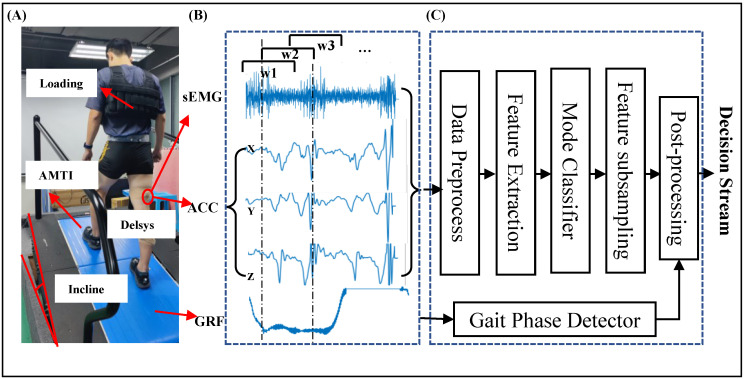
Experiment protocols. (**A**) Experiment setting (**B**) Data windowing scheme (**C**) Architecture of the neuromuscular–mechanical fusion system for locomotion mode and task recognition.

**Figure 2 bioengineering-11-00150-f002:**
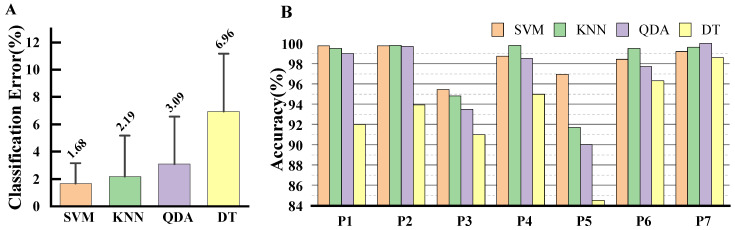
Classification performance of classifiers and participants. (**A**) The classification performance of SVM, KNN, QDA, and DT. Error bars in the graph represent the 95% confidence interval. (**B**) The recognition accuracy of four classifiers for participant 1–7. The accuracy rate is equal to 100% minus the classification error rate.

**Figure 3 bioengineering-11-00150-f003:**
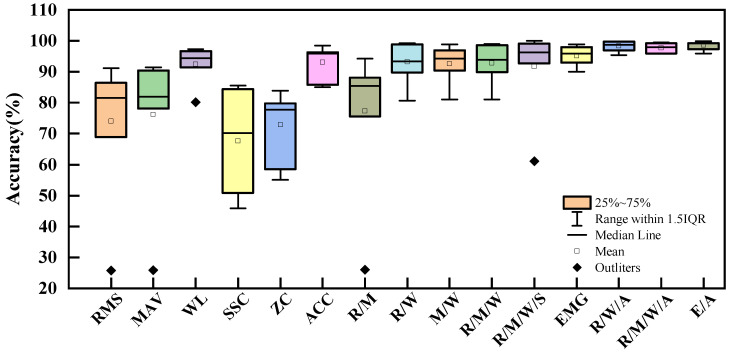
Influence of sEMG features and feature combinations on classification accuracy of SVM with 140 length of analysis window and 10 window increment.

**Figure 4 bioengineering-11-00150-f004:**
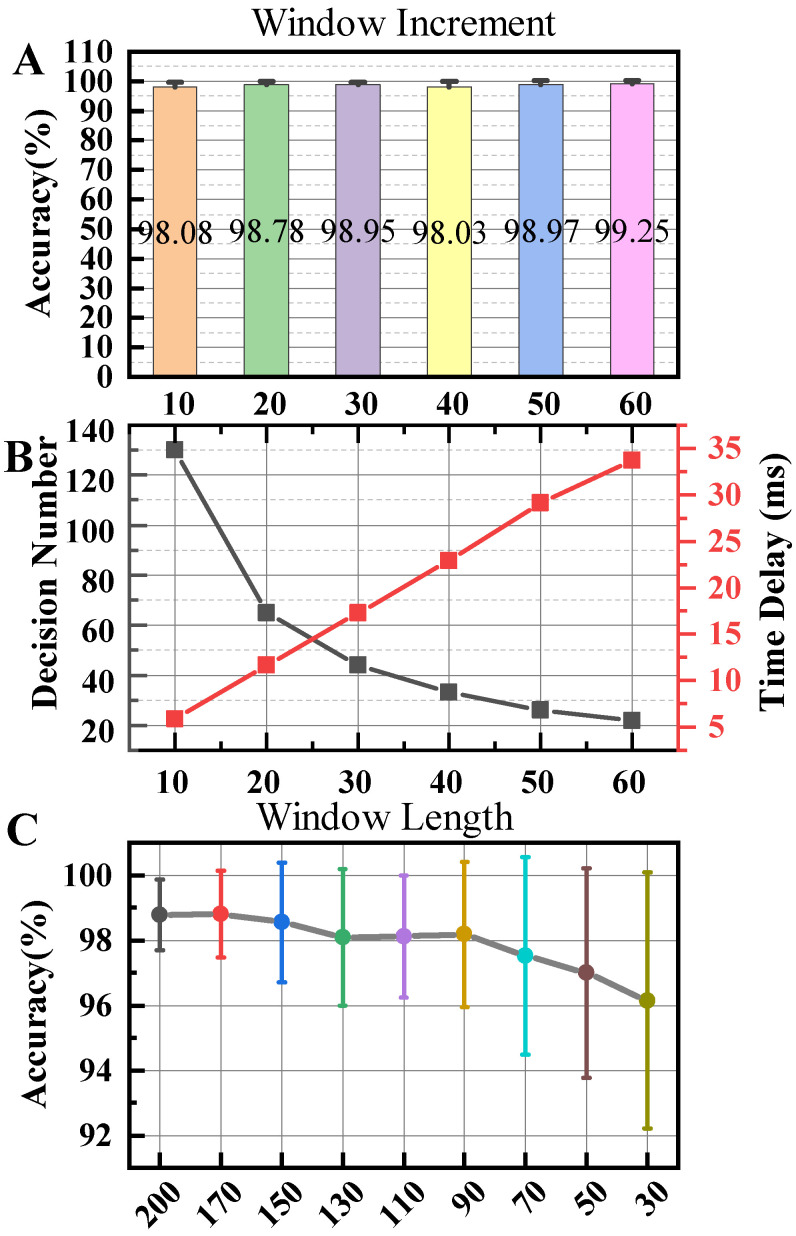
The influence of increment and length of analysis window of SVM with R/W/A feature set. (**A**) The effect of window increment on classification accuracy. Error bars in the graph represent the 95% confidence interval. (**B**) The black line represents the effect of window increment on decision number in one gait cycle. The red line represents the time delay between two decision numbers. (**C**) The effect of window length on classification accuracy. Error bars in the graph represent the 95% confidence interval.

**Figure 5 bioengineering-11-00150-f005:**
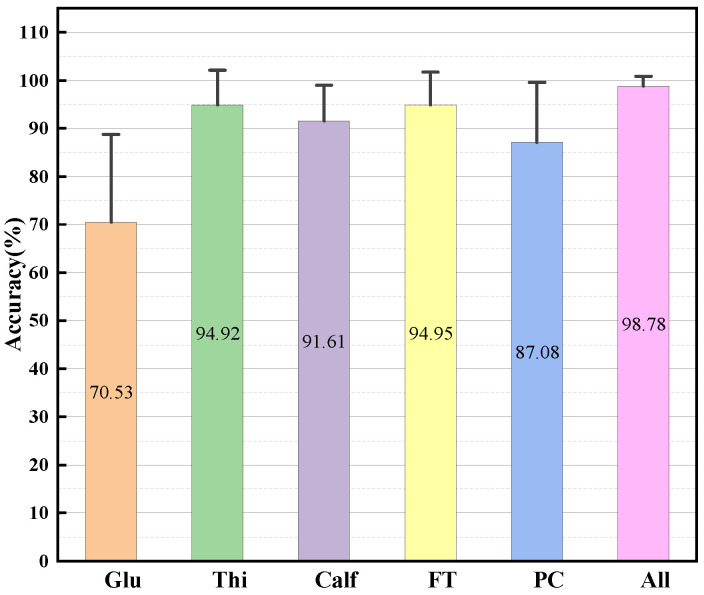
The robustness of the system. Error bars in the graph represent the 95% confidence interval. Glu = gluteus; Thi = thigh; AT = anterior thigh; PC = posterior calf. The components of each muscle group are shown in Table 3.

**Figure 6 bioengineering-11-00150-f006:**
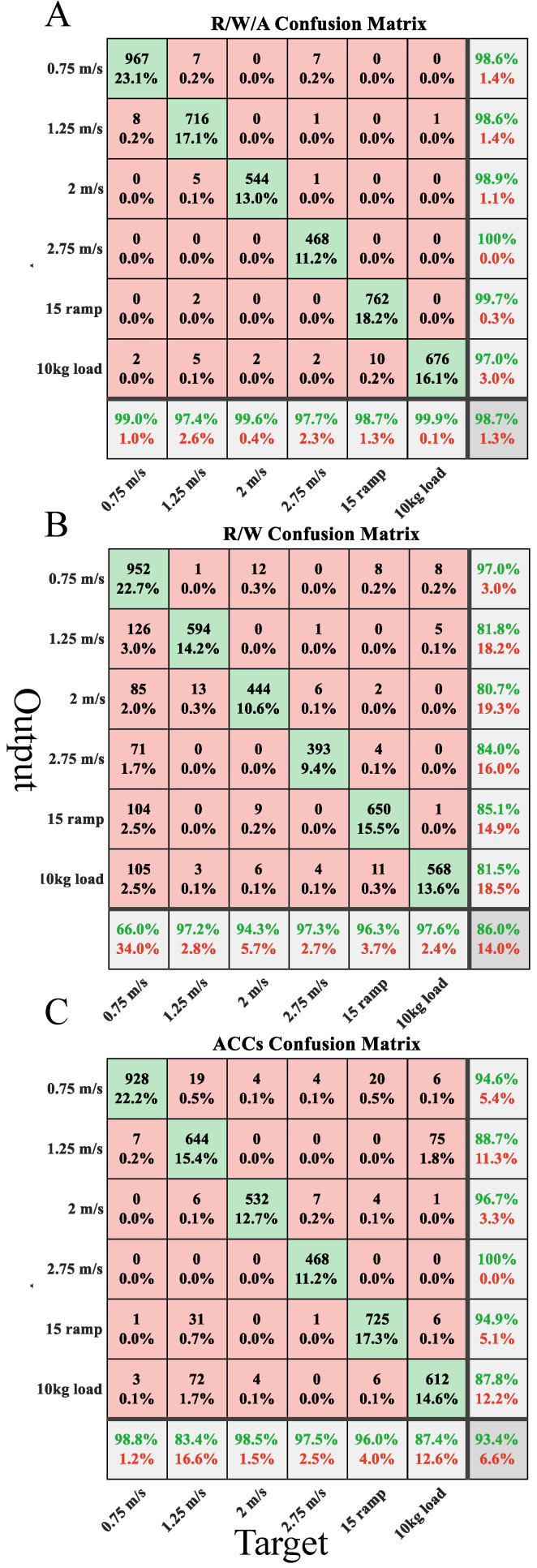
Confusion matrix for SVM classifier with 170 window length and 20 increment. The confusion matrix from (**A**–**C**) is R/W/A, R/W, and ACCs.

**Figure 7 bioengineering-11-00150-f007:**
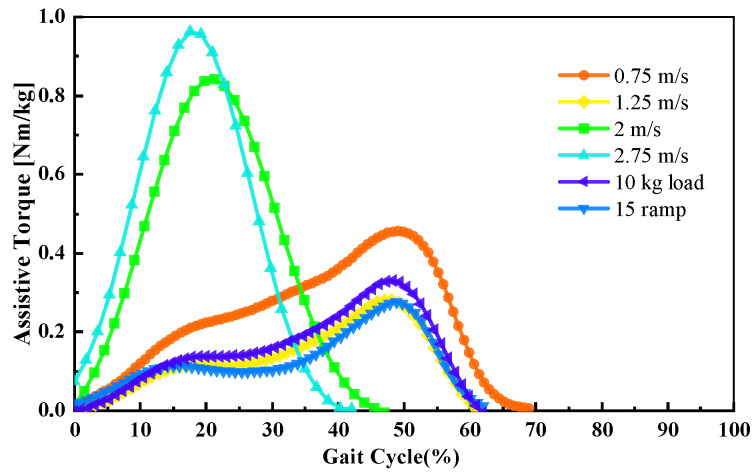
Continuous locomotion mode and task identifications with assistive torque decision results for P1.

**Table 1 bioengineering-11-00150-t001:** Anthropometric and demographic information of participants.

	Age	Weight (kg)	Height (cm)	Leg Length (cm)
P1	23	67.30	173	80
P2	24	61.70	174	88
P3	23	64.50	175	86
P4	23	72.65	174	83
P5	23	67.60	180	95
P6	26	66.70	173	84
P7	26	71.00	180	89

**Table 2 bioengineering-11-00150-t002:** Abbreviation and explanation of feature sets used in locomotion mode and task recognition.

Feature Set	Components	Dim of 1 Channel
ACC	ACCs	3
RMS	RMS	1
MAV	MAV	1
WL	WL	1
SSC	SSC	1
ZC	ZC	1
R/M	RMS, MAV	2
R/W	RMS, WL	2
M/W	MAV, WL	2
R/M/W	RMS, MAV, WL	3
R/W/A	RMS, WL, ACC	5
R/M/W/S	RMS, MAV, WL, SSC	4
R/M/W/A	RMS, MAV, WL, ACC	6
EMG	RMS, MAV, WL, SSC, ZC	5
ALL	RMS, MAV, WL, SSC, ZC, ACC	8

ACC = accelerations in x, y, and z directions; RMS = root mean square; MAV = mean absolute value; WL = waveform length; SSC = number of slope sign changes; ZC = number of zero crossings.

**Table 3 bioengineering-11-00150-t003:** Muscle groups and components used in the investigation of system robustness.

Muscle Group	Components	Number in Total
Gluteus	GM	1
Thigh	RF, VL, VM, BF	4
Calf	TA, GAS, SOL	3
Anterior Thigh	RF, VL, VM	3
Posterior Calf	GAS, SOL	2

GM = gluteus maximus; RF = rectus femoris; VL = vastus lateralis; VM = vastus medialis; BF = biceps femoris; TA = tibialis anterior; GAS = gastrocnemius medialis; SOL = soleus.

**Table 4 bioengineering-11-00150-t004:** The influence of ACCs on classification accuracy based on SVM with 140 length of analysis window and 10 window increment.

Participant	R/W	R/W/A	R/M/W	R/M/W/A	EMG	E/A
P1	99.28	99.73	99.00	97.92	96.37	99.27
P2	93.36	99.75	93.87	99.50	95.88	99.24
P3	98.77	95.46	96.68	96.25	97.90	99.30
P4	98.80	98.71	98.61	99.26	98.89	99.91
P5	91.77	96.94	89.92	95.97	90.00	95.97
P6	89.80	98.44	90.58	95.87	94.42	97.37
P7	80.62	99.19	81.03	99.10	93.00	98.94
Mean	93.20	98.32	92.81	97.70	95.21	98.57

**Table 5 bioengineering-11-00150-t005:** The window increment, minimum decision number, and maximum time delay in participants.

	10	20	30	40	50	60
P1	99.73 (119)	99.64 (60)	99.73 (40)	100 (30)	100 (24)	100 (20)
P2	99.75 (139)	99.50 (70)	99.25 (47)	99.00 (35)	99.58 (28)	100 (24)
P3	95.46 (109)	99.13 (55)	98.96 (37)	95.50 (28)	99.14 (22)	100 (19)
P4	98.71 (125)	99.45 (63)	99.72 (42)	99.26 (32)	98.89 (25)	99.91(21)
P5	96.94 (147)	96.63 (74)	98.07 (49)	95.97 (37)	90.00 (30)	95.97 (25)
P6	96.82 (143)	97.62 (72)	97.62 (48)	95.87 (36)	94.42 (29)	97.37 (24)
P7	99.19 (133)	99.51 (67)	99.27 (45)	99.10 (34)	93.00 (27)	98.94 (23)
Mean (Min)	98.08 (109)	98.78 (55)	98.95 (37)	97.70 (28)	95.21 (22)	98.57 (19)
Delay (ms)	5.88	11.65	17.32	22.88	29.12	33.71

**Table 6 bioengineering-11-00150-t006:** The influence of window length on classification accuracy.

	200	170	150	130	110	90	70	50	30
S1	99.64	100	100	100	100	100	100	99.33	99.00
S2	99.5	99.50	99.18	98.55	98.55	99.52	99.21	98.28	98.14
S3	99.13	100	100	100	100	99.51	98.86	98.87	97.60
S4	99.45	99.64	99.46	98.24	98.24	98.95	99.31	98.98	98.99
S5	96.63	96.98	94.33	93.97	93.97	92.96	90.44	89.47	86.89
S6	97.62	96.55	97.98	97.71	97.71	97.88	97.00	96.14	95.73
S7	99.51	98.88	98.89	98.43	98.28	98.45	97.86	97.88	96.69
Mean	98.78	98.79	98.54	98.08	98.11	98.18	97.52	96.99	96.15

## Data Availability

The data that support the findings of this study are available on request from the corresponding author.

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
