# Peer review of "Continuous Locomotion Mode and Task Identification for an Assistive Exoskeleton Based on Neuromuscular–Mechanical Fusion"

_bioengineering, 2024, doi:10.3390/bioengineering11020150_

Round 1
Reviewer 1 Report
Comments and Suggestions for Authors
This manuscript is suitable to be published at Bioengineering journal. The content is important for understanding locomotion mode for the development of assistive exoskeleton device in the future. However, minor corrections should be done by the authors before it can be accepted for publication.
1. Abstract
a) Specify the method in a single paragraph. At the moment, too general.
b) Conclusion is not clear. The last paragraph is only a hope, not a conclusion of the study. Please revise.
2. Introduction
a) Line 63 – typo for Figure? Or missing information here?
b) Research gap is unclear. What is the drawback? What is the problem from the literature?
3. Materials and Methods
a) I suggest including the consent form as an appendix.
b) Format for the consent form, is it from a standard?
c) How you determine the subject? Selection of subjects must be clear, why they have been chosen. Please state clearly here.
d) Procedure of anthropometric should be clearly explained. How and what equipment?
e) Eight muscle was selected. Which standard or reference?
f) Line 119 – the aim is to design an algorithm. So where is the design method? Please include. Need to explain further about it. This is a crucial part of the manuscript and it is a novelty of the study.
g) Line 136 – figure 1? It is better to put part a or b in the figure. Seems confusing.
4. Results
a) For me, this is excellent.
5. Discussion
a) Very limited discussion about the results and comparison between the author’s results with previously published literature by others. Authors should compare, at least find the difference between the stress from others and their own.
b) Should include limitation of study. What can be improved in the future? Suggest a minimum 1 paragraph.
6. Reference
a) Need to add more references. Especially for the discussion part.
Comments on the Quality of English LanguageSome grammatical errors found. Please correct it
Author Response
Dear Reviewer,
Thank you very much for reviewing our manuscript and providing valuable comments and suggestions. We sincerely appreciate your thorough review and detailed feedback. Please find below our responses and revisions based on your comments.
Yao Liu

Reviewer 2 Report
Comments and Suggestions for Authors
With few exceptions the content is fine, and and I am eagerly awaiting the results of the practical application, but there is an amount of formal things to give the contribution the quality it earns to have when published.
General remark: even AI has ist deep roots in the 20th century, and the scientific concepts of walking started in 1836. Thus, it is surprising that (as far as I found) your oldest reference is from 2006 (and partly references are meta-citations), and most are from Chinese authors. Why do you as scientists restrict your view onto world and history? Due to the value of your work, I renounce to make this a monitum in my review.
Exception #1: „verification“ in the abstract. Following the theory of science, you can not verify hypotheses, just refute them. And if you are able to refute them, you may believe them to be true until they are refuted. I think you mean „experimetal validation of the method“.
Exception #2: L. 187 – What is „a wide sence Gaussian random process“? And which is the objectve base of the statement „can be characterized as a wide sence Gaussian random process“?
In the whole manuscript, a lot of blanks is missing (but not consequently, thus you have to unify completely):
- - Separation of citation from the text: example [1]
- - In physical quantities, the space replaces the multiplier: 200 m/s
- - In the text: e.g. line 38 electromyography (sEMG)
L. 63: the number of figure is missing
L. 113: the product is named Trigno®
L. 116 following, L. 204 and Abbreviations: Following the international Terminologia anatomica, the names of musles are following the pattern M. glutaeus max., written in italics (if you are able to choose that style, here I am not) –> M. gluteus max. is tolerated, and in any case the subcase (max., med. for medialis) is abbreviated.
L 118 and Abbreviations: I guess GAS addresses M. gastrocnemius med.
L 128: why do you repeat the whole series from L. 121/122 (redundant information)? A reference like „listed above“ would be sufficient.
L. 201: „Amount of muscles. The amount of muscle …“ – What does this mean? Why the incomprehensible term additionaly as a buzzword in advance?
L. 269: „set, the“
L. 337: „Compared“
L. 362: In the PDF, the right text boarder is violated
Tab. 4 to 6: The annotation below should have a greater distance to the table
Tab. 5 and 6: like in Tab. 4 „No significant …“ as a statement is enough.
Tab. 6: „level“ and „0.05“ are disrupted.
Fig. 5: „Glu“ instead od of „Clu“
Author Response
Dear Reviewer,
Thank you for taking the time to review my book and provide your valuable feedback. I greatly appreciate your thoughtful comments and constructive criticism.

Reviewer 3 Report
Comments and Suggestions for Authors
It appears that this kind of research is warranted. Information is needed to clarify many areas in the methods and results/discussion. An expert English grammar review is needed for grammar, word choice, word tenses, punctuation, etc. I have made specific comments in the PDF.

quality of English is only fair
Author Response
Dear reviewer,
We are greatful for your comments and professional advice. These opinions help to improve academic rigor of our article. Based on suggestion and "request, we have made corrected modifications on the revised manuscript. We hope that our work can be improved again.
Yao Liu

Round 2
Reviewer 3 Report
Comments and Suggestions for Authors
The authors have made some effort to improve their manuscript. Yet, there are still major grammar, word choice, and word tense errors. In addition, clarification, ie, more information is needed in the methods section. Coherence (organization) in methods and results needs to be significantly improved. I did not review the discussion thoroughly until the other sections are improved. Specific comments can be found in the pdf.

Noted above. If there was an English grammar edit done for this revision it was lacking in quality and thoroughness.
Author Response

(The authors gave the same response as above.)

Round 3
Reviewer 3 Report
Comments and Suggestions for Authors
This revision represents a major effort and improvement. However, there are still several issues that need to be addressed. See specific comments in the pdf.

Much improved, but still with need for error corrections.
Author Response
Dear Reviewer,
Thank you so much for taking the time to review my paper. Your feedback and suggestions have been incredibly helpful and valuable to me. As this is my first paper, I apologize for any inconvenience caused. I truly appreciate your meticulous modifications and suggestions, which have had a great impact on my future writing papers. Once again, thank you for your time, effort, and valuable feedback.
Sincerely,
Yao Liu